# The Interplay Between Nutrition and Microbiota and the Role of Probiotics and Symbiotics in Pediatric Infectious Diseases

**DOI:** 10.3390/nu17071222

**Published:** 2025-03-31

**Authors:** María Slöcker-Barrio, Jesús López-Herce Cid, María José Solana-García

**Affiliations:** 1Pediatric Intensive Care Department, Hospital General Universitario Gregorio Marañón, 28009 Madrid, Spain; pielvi@hotmail.com (J.L.-H.C.); mjsolana@hotmail.com (M.J.S.-G.); 2Primary Care Interventions to Prevent Maternal and Child Chronic Diseases of Perinatal and Developmental Origin Network (RICORS-SAMID], RD24/0013/0012, Instituto de Salud Carlos III, 28029 Madrid, Spain; 3Gregorio Marañón Biomedical Research Institute, 28009 Madrid, Spain; 4Mother and Child and Public Health Department, School of Medicine, Universidad Complutense de Madrid, 28040 Madrid, Spain

**Keywords:** microbiota, nutrition, infection, immune system, probiotic, symbiotic, children

## Abstract

The interplay between nutrition and infectious diseases has been a central theme in health sciences for the last decades due to its great impact on the pediatric population, especially in immunocompromised patients and critically ill children. As conventional treatment and the development of antimicrobials for most infections standard treatment is either limited or not possible, alternative treatment options should be explored. Recent research shows that early enteral nutrition and nutritional supplements (such as probiotics and symbiotics) could have a pivotal role in promoting a healthy microbiome and subsequently preventing and improving outcomes for certain pediatric infectious diseases. However, understanding the specific mechanism of action and tailoring nutritional interventions remains a significant challenge. The optimal dose range for different probiotic strains and prebiotics and the most effective combination for each treatment indication needs further investigation and is yet to be defined. Additionally, in the era of personalized medicine, goal- and patient-directed treatment are key to optimizing and improving outcomes and minimizing potential complications and side effects, especially in complex and immunocompromised patients. The main objectives of this narrative review are 1. to explore the relationship and the complex interactions between microbiota and the human immune system; 2. to describe the influence of nutrition on infectious diseases; 3. to evaluate the impact of supplementation with probiotics and symbiotics in the prevention and treatment of the most relevant infections in children; and 4. to identify knowledge gaps and potential research priorities regarding the use of these supplements in pediatric patients.

## 1. Introduction

Nutritional support is one of the key pillars to avoid undernourishment in hospitalized patients, with enteral nutrition (EN) being the preferred method for nutrient delivery [1,2,3]. Moreover, EN is the first-line therapy in the treatment of certain diseases, such as Crohn’s disease [4], because it favors gastrointestinal mucosal trophism, prevents bacterial overgrowth and translocation, is less expensive, and is associated with a lower risk of infection compared to parenteral nutrition [5,6].

Infectious diseases are a leading cause of morbidity and mortality, especially in high-risk populations such as immunocompromised patients, children, or the elderly. The immune system, which is regulated by gut microbiota, plays a crucial role in the susceptibility, persistence, and clearance of these infections [7], highlighting the importance of a healthy microbiome.

The interplay between nutrition and infectious diseases has been a central theme in health sciences for decades. EN decreases infectious morbidity, as shown by previous randomized controlled trials (RCTs) in critically ill patients [8,9,10,11]. This effect is time and severity dependent, meaning that the beneficial effects appear when EN is initiated within the first 48 h of admission and in the sickest patients [3,8,9]. A prospective multicenter study of 207 critically ill patients showed that successful EN was associated with a reduction in infectious complications, particularly after 96 h of Intensive Care Unit (ICU) admission [10]. Although data regarding the relationship between the quantity of EN and infectious disease are limited, recent findings suggest that higher caloric and protein intakes may be associated with fewer infectious complications [10].

In the current era, where the role of microbiota is increasingly recognized as a critical component of human health, this relationship takes on new dimensions. The microbiota, comprising trillions of microorganisms residing in the human body, acts as a mediator between dietary inputs and immune responses, protects the gut barrier, and regulates metabolism as well as nutrient and drug absorption [11,12,13,14]. Understanding this triadic interaction between nutrition, microbiota, and infectious diseases can pave the way for innovative therapeutic and preventive strategies [15,16].

## 2. The Role of Nutrition in Shaping Microbiota

The human microbiome is composed of bacteria, viruses, fungi, and protozoa, and it changes throughout life according to several factors [17,18]. Previous studies suggest that the microbiome is mainly established during the first year of life, although other factors—such as genetic background [19], ethnicity or geographical location [20], early antibiotic exposure [17,21], and diet and lifestyle [22]—also contribute to its diversity.

Gut microbes play a crucial role in many aspects of human health including immune, metabolic, and neurobehavioral functions [13]; therefore, maintaining an equilibrated ecosystem is imperative. A healthy microbiota is mainly composed of Firmicutes and Bacteroidetes—which include genera such as *Lactobacillus*, *Bacillus*, *Enterococcus*, *Ruminococcus*, and *Clostridium* in the first group and Actinobacteria in the second group [23]. Another characteristic of a healthy gut is high alpha diversity, which indicates the number and distribution of different species within one sample, while beta diversity reflects differences in microbial composition between samples [24].

Diet is a primary determinant of microbiota composition and functionality; different dietary patterns can lead to changes in the microbiota profile. The amount, type, and balance of the main dietary macronutrients (carbohydrates, proteins, and fats) have a great impact on the large intestinal microbiota [25]. Generally, obesity and overweight are associated with lower diversity and a decrease in *Bacteroidetes* and *Akkermansia*, which can lead to gut barrier failure and a heightened inflammatory response [23,26].

Nutritional components, such as carbohydrates, fiber, fats, and proteins, influence microbial diversity and the production of bioactive metabolites like short-chain fatty acids (SCFAs) (e.g., butyrate, acetate, and propionate), which are very important for the human body [15]. Butyrate is the main energy source for colonocytes, maintaining glucose and energy homeostasis and favoring apoptosis in malignant colonic cells; propionate regulates satiety and gluconeogenesis; and acetate is used in cholesterol metabolism, lipogenesis, and may also be involved in appetite regulation [13]. The metabolic pathways of microbiota are represented in Figure 1.

Fat quantity and saturation influence gut microbiota. Monounsaturated and medium-chain fatty acids are beneficial because they improve microbiota equilibrium and diversity, thereby enhancing gut barrier function, lipid metabolism, and cognitive and metabolic functions [23]. However, a high-fat diet rich in saturated fatty acids—with high omega-6 PUFAs and low omega-3 PUFAs—can lead to dysbiosis and increased gut barrier permeability, potentially predisposing individuals to infections and endotoxemia [27].

Non-digestible carbohydrates, also known as fiber, are essential substrates for maintaining a healthy microbiota because they promote the proliferation of beneficial bacteria, like Bifidobacteria and Lactobacilli, which are involved in colonic fermentation and increasing SCFA production that also has anti-inflammatory properties [26,28]. Conversely, a high-sugar diet and low levels of microbiota-accessible nutrients [such as inulin and oligofructose] increase the levels of harmful bacteria that produce enzymes degrading the mucous layer, thereby increasing gut permeability [29,30].

The quantity and quality of proteins also contribute to the equilibrium of the microbiota because, depending on the type of protein consumed, by-products, such as SCFAs, branched-chain fatty acids, or toxic substrates, like ammonia or nitrosamines, can be formed [25]. Consumption of plant-based proteins, such as those from pulses or peas, is associated with a better microbiota composition and higher levels of beneficial microorganisms (e.g., *Lactobacillus*, *Bifidobacterium*, *Clostridium*, *and Roseburia*) that induce the production of acetate and butyrate, thereby reducing gut inflammation [31,32]. Faba beans have also shown beneficial properties because they contain bioactive peptides, phenolic compounds, GABA, and L-DOPA with antidiabetic, antihypertensive, cholesterol-lowering, and anti-inflammatory effects [33]. In contrast, diets enriched in animal proteins favor the growth of *Alistipes*, *Bilophila*, and *Bacteroides* and the formation of trimethylamine-N-oxide, secondary bile acids, SCFAs, and aromatic amino acids—all of which are implicated in cardiovascular disease—as well as an increase in Desulfovibrio, which may be related to gut inflammation [34,35,36,37].

Micronutrient deficits can also affect the equilibrium of microbiota. Deficiencies in vitamins A, D, and zinc impair immune function and disrupt microbial homeostasis, heightening the risk of infections [38].

EN seems to shape the microbiota. Although the mechanism has not been well established, it has been proposed that EN can disrupt dysbiotic communities and replace them with a “healthier microbiota” [39].

A report conducted on a child with chronic granulomatous disease showed that exclusive enteral nutrition induced significant changes in the gut microbiota by increasing alpha diversity and favoring the growth of *Faecalibacterium prausnitzii*, *Dialister propionicifaciens*, and *Parabacteroides merdae* while reducing the abundance of certain pathogenic species [40]. In critically ill patients, successful enteral nutrition has been associated with a reduction in infectious complications, particularly after 96 h of ICU admission [10]. Although microbiota were not analyzed in that study, the beneficial outcomes related to increased caloric and protein intake may have contributed to improving microbial equilibrium, leading to better patient outcomes.

The effect of gut microbiota on nutritional tolerance in critically ill patients remains unclear; however, recent evidence suggests that microbiota could be involved [41]. An observational study in non-abdominal septic ICU patients showed that microbiota characteristics—such as higher alpha diversity and increased levels of *Parabacteroides*, *Bacteroides*, and *Bifidobacterium*—were associated with better enteral nutrition tolerance [42]. This may be explained by the role of the intestinal flora and its metabolites in digestion, intestinal motility, and immune function.

Moreover, EN modifies the composition of the microbiota in sick patients and healthy populations. A study analyzing the impact of EN on the microbiota in newly diagnosed Crohn’s disease patients and healthy children found significant microbial changes after EN initiation, with alterations occurring even more rapidly in healthy siblings [43]. A study analyzing the impact of EN on microbiota in newly diagnosed Crohn’s disease and healthy children showed that both groups showed significant microbiota changes after EN onset and that these changes appeared even more rapidly in healthy siblings [43]. Notably, changes in gut microbiota may persist for up to two months after EN cessation without a complete return to the previous composition [41].

## 3. Nutrition, the Immune System, and Infectious Diseases

The gut microbiota, alongside the mucosal immune system and epithelium, represents one of the main hurdles that pathogens must overcome to cause an infection. The gut microbiota plays a crucial role in the development and modulation of the immune system by educating immune cells and regulating the balance between pro-inflammatory and anti-inflammatory responses. In addition, the immune system interacts with the digestive system, where micronutrients and gut microbiota modulate inflammatory and immunoregulatory processes—including the regulation of mucosal immunity in the airways to protect against respiratory infections [44,45].

Recent research has revealed that the gut microbiome not only regulates local mucosal immunity but also shapes systemic immune responses [18,30,46,47,48,49]. By releasing soluble microbial products into circulation, microbiota influence the activation and differentiation of various immune cells, including promoting the development of regulatory T cells via short-chain fatty acids, such as butyrate. This dynamic interplay modulates both adaptive and innate immune functions—affecting T-helper cell differentiation, B cell regulation, macrophage activation, and natural killer cell maturation—thereby maintaining immune homeostasis and protecting against systemic inflammation [7]. Additionally, the gut–lung axis highlights the microbiome’s impact on respiratory health, as alterations in gut microbial balance have been linked to an increased risk of allergic airway diseases and respiratory infections [7].

Dysbiosis, or an imbalance in microbial communities, can compromise this regulation, leading to increased susceptibility to gastrointestinal infections, such as *Clostridioides difficile* and *Salmonella* [50], and systemic effects that heighten vulnerability to respiratory infections [16], including influenza and COVID-19.

Insufficient intake of energy and protein significantly impairs both innate and adaptive immune functions [46,51]. Immune cells depend on various biochemical pathways to obtain the energy and metabolites they need, and malnutrition disrupts these processes—hindering cell proliferation, metabolism, and differentiation. Infectious diseases can significantly alter nutritional status by increasing metabolic demands while reducing nutrient absorption, leading to deficiencies that further weaken immune defenses. In fact, during infections, the immune system may require up to 25–30% of basal metabolic energy, meaning that low-energy intake compromises its ability to respond effectively. Additionally, antibiotic treatments for infections frequently disrupt the microbiota, causing secondary issues, such as diarrhea and an increased risk of opportunistic infections [38].

Evidence of the effects of insufficient nutritional intake has been observed in different populations [47,48]. For example, elite athletes experiencing low-energy availability, resulting in a 13% weight loss, showed signs of immunosuppression—including dysregulated hematopoiesis, reduced immune cell proliferation, and diminished antibody and chemokine secretion [47]. Similarly, the demanding physical conditions and energy restrictions typical of military training have been associated with increased susceptibility to infections due to suppressed immune function [48].

Protein status is closely linked to cell-mediated immunity; protein-energy malnutrition impairs immune function by worsening epithelial and physiological barrier integrity, degrading the function of macrophages, T lymphocytes, neutrophils, and natural killer cells and causing lymphoid organ atrophy—all of which increase susceptibility to viral and bacterial pathogens and opportunistic infections [49,51,52].

Micronutrients play a crucial role in immune function. Insufficient EN intake leads to micronutrient deficits that can increase the risk and severity of infections. Selenium, incorporated as selenocysteine in selenoproteins, is crucial for antioxidant defense, redox signaling, and maintaining redox homeostasis during viral infections. A deficiency results in decreased selenoprotein expression, increased oxidative stress, viral genome mutations, and heightened pathogenicity [e.g., in Hepatitis B, Hepatitis C, and influenza] [53,54,55]. Beyond regulating mineral metabolism, vitamin D exerts immunomodulatory effects by enhancing epithelial barrier integrity and upregulating antimicrobial peptides such as cathelicidin. Low vitamin D levels have been linked to an increased risk of respiratory infections and longer hospital stays [56,57,58,59]. Vitamin C supports the growth and function of immune cells, maintains epithelial barrier integrity, and protects against oxidative damage during pathogen clearance—a role that becomes especially critical for ICU patients who may have increased requirements [60,61,62,63,64]. Iron is vital for both innate and adaptive immunity and oxygen transport; however, during periods of stress and inflammation, elevated hepcidin levels can impair iron absorption, thereby compromising immune function [65]. Finally, zinc is essential for the development and maintenance of immune cells and supports gut mucosal immunity by limiting parasite survival and modulating systemic responses. Insufficient zinc status is associated with increased susceptibility to viral infections and can be exacerbated by inflammation [56,66,67,68].

### Malnutrition, Gut Microbiota, and Infectious Diseases in Children

During childhood, the transition from breastfeeding to solid food intake plays a crucial role in the maturation of the gut microbiota, establishing a stable metabolic state and highlighting the importance of nutrition in microbiota development [69,70,71].

Malnutrition—including both undernutrition and overweight—is a metabolic condition caused by an imbalance between nutrient intake and the requirements necessary for growth and metabolism [72]. This imbalance leads to dysbiosis [73].

Undernutrition is the most common form of malnutrition during childhood [74]. It is particularly significant in this population due to its severe cognitive, physical, and neurological consequences, as well as its association with high mortality rates [75].

Among the multiple factors contributing to undernutrition, intestinal infections play a major role [76]. Enteric pathogens damage the intestinal mucosa and epithelial cells, leading to diarrhea and malabsorption of carbohydrates and other essential nutrients, thereby promoting weight loss and malnutrition. In addition, children suffering from diarrhea may experience changes in their gut microbiota, with an increased abundance of Fusobacterium and a higher relative abundance of Bacteroides [77].

A recently identified condition, environmental enteric dysfunction (EED), may also affect gut microbiota composition. EED is a subclinical disorder of the small intestine characterized by villous atrophy and crypt hyperplasia, resulting from chronic exposure to environmental pathogens in settings with poor sanitation [78]. It is hypothesized that EED alters the structure and function of gut microbiota, ultimately contributing to stunted growth.

Overnutrition is likewise associated with dysbiosis, typically characterized by reduced microbial diversity, an increased proportion of Firmicutes, and a decreased proportion of Bacteroidetes [25]. In these individuals, microbial imbalance is linked to insulin resistance and altered metabolic activity in peripheral tissues.

Malnutrition itself can disrupt the balance of the gut microbiota in children [72]. Previous studies have demonstrated that malnourished children exhibit reduced microbial diversity, increased Proteobacteria, and decreased levels of Bacteroides, Bifidobacterium, and Lactobacillus compared to healthy peers [76]. The depletion of Bifidobacteria often appears as the initial change, followed by colonization by potentially harmful bacteria such as *Escherichia coli*, *Streptococcus* spp., and *Fusobacterium mortiferum* [79]. This dysbiotic microbiota also contributes to recurrent episodes of diarrhea.

This condition may be further aggravated by the low concentration of SCFAs and the release of pathogenic components, like trimethylamine N-oxide or gamma-aminobutyric acids, from the gut microbiota [73].

The interaction between nutritional status and the gut microbiota represents a promising area for future research

## 4. The Role of Probiotics, Prebiotics, and Symbiotics in Pediatric Infectious Diseases

### 4.1. Definition and Mechanism of Action of Probiotics, Prebiotics, and Symbiotics

In 2014, an expert group gathered by the International Scientific Association for Probiotics and Prebiotics (ISAPP) defined probiotics as “live microorganisms which when administered in adequate amounts confer a health benefit on the host” and prebiotics as “a non-digestible fermented substrate that is selectively used by host microorganisms conferring a health benefit” [80,81]. A symbiotic would be defined as the synergistic combination of both probiotics and prebiotics with the goal of modulating not only the microbiota but also the metabolic activity in the gut [80,81]. The ISAPP also recognized two subgroups of symbiotics, synergistic (the prebiotic is meant to be used only by the coadministered microbial strains) and complementary [the prebiotic effect is aimed at the patient’s resident microbiota] [82].

These microorganisms include certain types of lactic acid bacteria (*Lactobacillus and Bifidobacterium*), the non-pathogenic strain of *Escherichia coli E. coli* Nissle, *Clostridium butyricum*, *Streptococcus salivarius*, *Enterococcus* spp., and the yeast *Saccharomyces boulardii* [83]. The number of probiotics is expressed by the number of viable cells or colony-forming units (CFUs). Most supplements contain 1 to 10 billion CFUs per dose, but multiple specific strain preparations and strain combinations can be found in the market [84]. Furthermore, the most frequently used prebiotics include fructooligosaccharides (FOSs) and galactooligosaccharides (GOSs) [82].

The mechanisms by which probiotics play a role in immune modulation are not completely understood. Previous research has shown that they can influence both innate and adaptive immune responses by interacting with intestinal epithelial cells, lymphocytes, dendritic cells, monocytes, and macrophages. They enhance the release of immunoglobulin A by B cells, thereby improving mucosal immunity and also the expression of interleukin (IL)-6, IL-10, and tumor necrosis factor (TNF)-β [85,86,87]. Probiotic products can interfere with the invasion of the gut endothelium by pathogens, improving the integrity of enterocyte tight junctions and also stimulating local mucin, lactic acid, and hydrogen peroxide production, which lowers intraluminal pH [84]. Specific effects can be attributed to certain probiotic strains. For example, *Saccharomyces boulardii* produces protease enzymes that promote the neutralization of toxins, such as those produced by *E. coli*, *C. difficile*, or *Vibrio cholerae*. On the other hand, *Lactobacilli* produce galactosidase, which can facilitate lactose digestion, thus preventing the development of diarrhea [88].

### 4.2. Treatment with Probiotics, Prebiotics, and Symbiotics in Pediatric Infectious Diseases

The use of these microbial strains and their combination with different oligosaccharides in pediatric infections has been an extensive field of research in the last two decades. This section will focus on the available evidence regarding the use of probiotics, prebiotics, and symbiotics in significant or prevalent infections affecting pediatric patients.

#### 4.2.1. Acute Viral Gastroenteritis (AGE)

Although many studies have shown the beneficial effects of probiotics in pediatric AGE and its administration has been recommended by clinical guidelines [89,90], there is still a lack of consensus and formal indications on their clinical use. This might be due to the great variety of available probiotic formulations, the functionality of each strain, the etiology of AGE, and the optimal dosage for each group. The most frequently used probiotics and combinations, along with the dose range described in different studies, are shown in Table 1. Most studies and systematic reviews have shown a reduction in the duration of diarrhea between 15 and 45 h and the length of hospital stay between 1 and 2 days for both viral (predominantly Rotavirus) and bacterial diarrhea in hospitalized patients treated with individual strains or different combinations of Bifidobacterium, Lactobacillus, and Saccharomyces species. Despite the mild reduction in symptom duration, the global repercussions of such a frequent diagnosis as AGE in hospitalized children could be considered significant [86,88,91,92]. However, two RCTs with over 800 patients performed in the United States and Canada found no benefit associated with probiotic administration (*L. rhamnosus GG* and the same strain combined with *L. helveticus*). These authors performed a separate analysis regarding viral etiology and concluded that potential benefits could be observed in rotavirus AGE but not in norovirus and adenovirus AGE, which might be predominant in settings with a high rotavirus vaccine coverage [93,94]. This supports the hypothesis that probiotics are associated with pathogen-specific benefits, with the greatest effect being observed in rotavirus AGE and more limited benefit in other viral and bacterial pathogens. Considering these data, a 2020 published Cochrane systematic review concludes that there is not enough evidence providing a pathogen-specific analysis of the role of probiotic use in AGE due to the small number of studies identifying etiological agents of diarrhea or conducting specific statistical analysis focused on individual pathogens and raises concerns about the safety and efficacy of different combinations and dosages of probiotic strains [88,95].

Regarding symbiotics, the European Society of Pediatric Gastroenterology, Hepatology, and Nutrition (ESPGHAN) Study Group on Gut Microbiota evaluated two systematic reviews and meta-analyses published by Yang et al. [96] and Vassilopoulou et al. [97] in 2019 and 2021 evaluating both probiotics and symbiotics. Globally, a total of six RCTs and high-quality studies evaluating symbiotics were evaluated. These papers evaluated different mixtures of probiotic strains (*Lactobacillus*, *Bifidobacterium*, *Streptococcus*, and *Lacticasei*) plus FOSs, GOSs, inulin, or zinc. The experts concluded that only one symbiotic preparation (*S. thermophilus*, *L. rhamnosus*, *L. acidophilus*, *B. lactis*, and *B. infantis* associated with FOSs) was evaluated in two RCTs, one of which lacked sufficient statistical power. Additionally, the difference between groups was approximately 1 day. Although clinical significance could be called into question, the 25–30% reduction in duration and costs of such a frequent disease could be clinically significant. In conclusion, there were no two adequate and well-designed studies evaluating the same symbiotic preparation, so the effectiveness of this intervention could not be determined and no recommendation could be formulated [82,96,97].

#### 4.2.2. Upper Respiratory Tract Viral Infections (URTIs)

Experimental research on the pathogenesis of viral respiratory infections (RTIs) has shown that microbial agents, including probiotics (*Lactobacillus*, *Enterococcus*, and *Bifidobacterium*) and prebiotics (FOS, GOS, and inulin), may influence the composition of gastrointestinal flora and provide beneficial effects for these patients, such as improving viral clearance and shortening disease duration. This positive effect has been attributed to the gut–lung axis theory by the regulation of antiviral defense gene expression in the airway’s macrophages and the enhancement of early antiviral response through interferon and other inflammatory cytokine signaling [98]. A meta-analysis published by Wang et al. [99] in 2021 analyzed 45 preclinical studies performed in mice, chickens, calves, and pigs. In these experimental studies, focused mainly on influenza and respiratory syncytial virus, the authors evaluated different symbiotic combinations of soluble oligosaccharides and lactic acid bacteria. The pooled data showed a decrease in both viral load and mortality, and the authors attributed this beneficial effect to the shift of the immune response in favor of the anti-inflammatory cytokines IFN-α and γ, IL-1β, and IL-12 and against the production of pro-inflammatory cytokines IL-6 and TNF-α [99].

The relationship between nasopharyngeal microbiota and the clinical course of respiratory infections has been explored by several authors. In a study recently published by Penela-Sánchez et al. in 104 children with Rhinovirus and Enterovirus RTIs, the quantity and diversity of nasopharyngeal microbiota were decreased among children with severe infection. These authors found that the genus Dolosigranulum was related to respiratory health, while the genus Haemophilus was specifically predominant in children with severe RV/EV LRTIs, suggesting a close relationship between the nasopharyngeal microbiota and different clinical presentations of infection [99].

The previously mentioned meta-analysis published by Wang et al. in 2021 [99] also analyzed six randomized double-blind and placebo-controlled trials. These were focused mainly on Rhinovirus and to a lesser extent on influenza and SARS-CoV-2 infection and included different combinations of probiotic strains (*Bifidobacterium lactis*, *Lactobacillus rhamnosus* and *acidophilus*, *Lactococcus lactis*, and *Streptococcus termophilus*) and prebiotics (GOSs, sucrose, and polydextrose) versus sucrose alone or placebo. These RCTs showed a reduction in the viral load, the incidence of infections, the symptom severity and need for ICU admission, and mortality. Two of the studies also observed an increase in anti-inflammatory cytokines and a decrease in pro-inflammatory cytokines. However, these clinical data failed to demonstrate statistical significance, probably due to the smaller sample size and considerable heterogeneity between the different RCTs compared to the experimental studies [99,100].

Another systematic review and meta-analysis of RCTs by Chan et al. in 2021 [101] evaluated the impact of symbiotics on preventing URTIs in more than 10.000 patients, including adults and children. Overall, symbiotic interventions reduced both the incidence rate and the proportion of RTIs amongst participants by 16%. A subgroup analysis suggested more prominent effects of symbiotics among adults than infants and children [101]. Nevertheless, Zhao et al. performed a similar approach with symbiotics and reported a larger risk reduction in URTIs (47%), whereas the effect was only significant in children but not in adults, and the episode rates of URTIs did not differ significantly between symbiotic and control groups [102]. Lastly, another systematic review and meta-analysis of RCTs by Williams et al. in 2022 [103] included 58 studies (mostly RCTs) examining the effect of prebiotics or symbiotics on the incidence, severity, or duration of RTIs and/or markers of immune function (e.g., immunophenotype or natural killer (NK) cell function). The meta-analysis pointed out that the numbers of subjects with one or more RTIs were reduced with prebiotic and symbiotic supplementation compared to placebo, and there was an improvement in innate immunity due to an increase in NK cell function [103].

The most frequently used probiotics and combinations, along with the dose range described in different studies for the prevention of respiratory tract infections in children, are shown in Table 2.

#### 4.2.3. Sepsis

Systemic inflammation can disrupt the gut environment, playing a key role in the dysregulated cascade that leads to organ dysfunction. Disruption of intestinal epithelial homeostasis during sepsis leads to a shift in microbiota favoring pathogens over beneficial bacteria, increased pro-inflammatory cytokine secretion, compromised barriers, and apoptosis, culminating in multi-organ failure. Understanding the impact of gut microbiota on sepsis pathophysiology is crucial. Experimental studies in mice suggest that a diverse and balanced gut microbiota can improve host defense against both enteric and systemic pathogens. Therefore, an imbalance in gut microbiota could make the host prone to the development of sepsis [104,105].

Recent RCTs have shown the potential of probiotics in both children and adult populations within the ICU. Considering probiotics and symbiotics as potential treatments could be a cost-effective preventive strategy, which is particularly interesting in an environment highly burdened by antimicrobial resistance [106].

Angurana et al. [107] conducted an RCT investigating the impact of probiotics on cytokine levels in critically ill children with severe sepsis, showing a marked reduction in IL-6, TNF-α, and C-reactive protein (CRP) levels after probiotic administration [96]. Similarly, recent research has also reported a significant decrease in CRP and IL-6 levels in critically ill patients after the administration of probiotics [108,109,110,111]. Currently, there are ongoing RCTs focused on investigating the effect of symbiotic supplementation in critically ill septic children admitted to the pediatric ICU (PICU), which will probably show promising results [105].

#### 4.2.4. Healthcare-Associated Infections, Antibiotic-Associated Diarrhea, and Infections Caused by Multidrug-Resistant Bacteria

Ventilator-associated pneumonia (VAP) and trachaeobronchitis are among the most frequent infectious complications in patients admitted to the ICU, alongside catheter-related bloodstream infections. These respiratory infections occur at least 48 h after the onset of mechanical ventilation (MV) and are an increased source of morbidity (prolonged duration of MV, PICU, and hospital length of admission), as well as increased mortality and healthcare costs [112,113]. Despite the decrease in incidence since the implementation and generalization of the VAP preventive bundles and the development of new-generation antibiotics, their repercussions on critically ill patients remain significant. Thus, there have been many studies trying to explore different non-antibiotic approaches for the prevention of VAP in the last few years. A systematic review and meta-analysis of 23 adult and children studies performed by Sun et al. in 2022 [112] concluded that the prophylactic administration of probiotics had a positive impact on the incidence of VAP, with a reduction in risk of 31% and 45% in adults and neonates/children, respectively, although these results in children have to be interpreted with caution, and further investigations in this area are warranted [112].

Antibiotic treatment is a known risk factor for intestinal dysbiosis and an imbalance and alteration of the gut microbiota, resulting in potentially severe diseases, such as antibiotic-associated diarrhea (AAD). Probiotics and symbiotics have proven efficacy in preventing and delaying the onset of AAD, and specifically, some studies have shown a beneficial effect of probiotics in AAD compared to the standard oral rehydration therapy in children [80,114,115]. A study performed in pediatric patients aged 3–6 years old taking antibiotics showed that prebiotics, such as inulin and other fructans, induced specific changes in the gut microbiota by increasing bifidobacteria levels, decreasing the occurrence of antibiotic-induced disturbances and producing beneficial effects in clinical outcome [116].

Antibiotic-resistant bacteria are a growing threat throughout the globe, and as new antibiotic drug development has failed to keep up with the spread of multidrug-resistant microorganisms (MDROs), the development of new treatment strategies focusing on both microbiome and resistome has become an emerging field of interest. A systematic review published in 2021 suggested that probiotics may play a beneficial role in critically ill patients, both in decreasing MDRO colonization and infection and thus decreasing the need for broad-spectrum antibiotic therapy, without significant adverse events. There are several studies planned or ongoing focused on the effects of probiotics (*Bifidobacterium*, *Lactobacillus*, and *Saccharomyces boulardii*) on MDRO colonization including vancomycin-resistant *Enterococcus*, carbapenem-resistant *Enterobacteriaceae* (CRE), extended-spectrum beta-lactamases (ESBLs), *Enterobacteriaceae*, and meticillin-resistant *Staphylococcus aureus* (MRSA) [117].

## 5. Future Directions and Challenges

Despite the significant advances and promising findings about microbiota and their interactions with the host environment and immune system, significant challenges remain. Interindividual variability—due to differences in genetics, lifestyle, previous health conditions, and baseline microbiota composition—makes it difficult to generalize nutritional interventions. In addition, a better mechanistic understanding is needed to elucidate the exact processes through which specific nutrients and microbiota modulate immune responses.

The field regarding the use of nutritional supplements such as probiotics, prebiotics, and symbiotics is also in constant development. Despite the existing research showing promising results, it is difficult to apply these findings to clinical practice for several reasons. First, the wide range of different probiotic strains and oligosaccharide formulations and the variation in both dosage and treatment duration proposed in the different studies make it difficult to compare the results and draw conclusions that can become the basis for actual recommendations applicable to clinical practice. Second, there are several studies focused on the positive effects of these nutritional supplements but fewer studies focused on assessing the risks and potential side effects of using probiotics and symbiotics in special and more complex pediatric populations such as premature babies, immunosuppressed, and critically ill patients. Although, in general, probiotics are considered safe, there has been a reasonable concern regarding their safety due to reported cases of bacteriemia, sepsis, endocarditis, and other local infections in these specific populations. There is also growing concern about the potential transfer of antimicrobial resistance genes from probiotic strains to pathogenic bacteria, which should also be addressed in upcoming investigations. Finally, in the new era of personalized medicine, developing scalable, cost-effective, and patient-directed interventions for diverse populations remains a critical goal.

As a limitation of this present research, the narrative review format offers a wide overview of the topic at hand, highlighting the most relevant evidence and the potential knowledge gaps. However, it does not take into consideration the full body of evidence that can be provided by other kinds of studies, such as systematic reviews or meta-analyses, which can offer a deeper insight and make conclusions more generalizable.

## 6. Conclusions

The nexus of nutrition, microbiota, and infectious diseases represents a fertile ground for advancing health sciences. By harnessing the power of both diet and supplements to modulate the microbiota—and consequently immune function—we can develop novel strategies to combat infectious diseases. Although evidence has grown considerably over the last few years in this promising field, there is a need to conduct more randomized clinical trials to find the optimal symbiotic combination/s, dosing, and treatment duration for every clinical indication. Moreover, it is necessary to perform these trials also in special populations to assess potential risks and avoid undesirable side effects. Collaborative efforts between nutritionists, microbiologists, and clinicians will be essential in translating this knowledge into practical applications that benefit global health.

## Figures and Tables

**Figure 1 nutrients-17-01222-f001:**
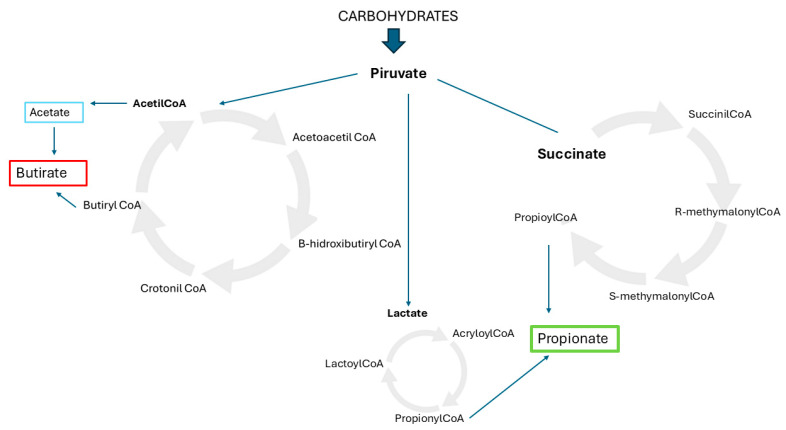
Metabolic pathways of microbiota.

**Table 1 nutrients-17-01222-t001:** Types of single and combined probiotic preparations for the treatment of children’s diarrhea and dose range.

Type of Probiotic Strain or Combination	Dosage
*Bifidobacterium lactis*	14.5 × 10^6^ CFU/100 mL milk for 7 days
*Lactobacillus rhamnosus* GG (LGG)	10^9^/day–10^10^ CFU/2× for 5 days
*Lactobacillus paracasei* ST 11	5 × 10^9^ CFU/2× for 5 days
*Lactobacillus reuteri* DSM 17938	10^8^ CFU
*Lactobacillus acidophilus*	10^8−9^ CFU/2× for 5 days
*Saccharomyces boulardii*	250–500 mg/day or 4 × 10^9−10^ viable cells for 5 days
*Escherichia coli* Nissle	10^8^ CFU/day–10^10−11^ CFU/for 5 days
*B. longum*, *B. lactis*, *L. acidophilus*, *L. rhamnosus*, *L. plantarum*, and *P. pentosaceus*	10^8^ CFU each strain
*L rhamnosus* 19070-2 and *L. reuteri* C+DSM 12246	1.7 × 10^10^ + 0.5 × 10^10^ CFU
LGG, *S. boulardii*, *B. claudii*, *L. delbrueckii var bulgaricus*, *Streptococcus thermophilus*, *L. acidophilus*, *B. bifidum*, and *Enterococcus faecium* SF68	10^7−9^ CFU each strain
*L. casei*, *L. rhamnosus*, *L. acidophilus*, *L. bulgaricus S. termophilus*, *B. breve*, and *B. infantis*	10^9^ CFU/2× for 5 days

CFU: colony-forming unit. 2×: twice daily. Adapted from Do Carmo et al. [92], Sasaran et al. [88], and Maftei et al. [84].

**Table 2 nutrients-17-01222-t002:** Types of single and combined probiotic preparations for the prevention of children’s respiratory tract infections and daily dose range.

Type of Probiotic Strain or Combination	Dosage
*Bifidobacterium lactis*	1–1.9 × 10^9^ CFU for 3 weeks to 12 months
*Lactobacillus rhamnosus* GG (LGG)	10^9^ CFU for 1–2 months
*Lactobacillus paracasei*, *L. plantarum*, and *B. lactis*	10^10^ CFU for each strain for 3 months
*L. rhamnosus*, *S. termophilus*, and *S. salivarius*	10^7^ CFU for each strain for 12 months
*L. acidophilus* and *B. lactis*	5 × 10^9^–1 × 10 ^10^ CFU for 2–8 weeks
*L. bulgaricus*, *Streptococcus thermophilus*, and *B. lactis*	10^8^ CFU for each strain for 16 weeks
*B. longum* and *S. termophilus*	10^7^ CFU for each strain for 3 months
*L. rhamnosus*, *L. plantarum*, and *B. lactis*	10^10^ CFU for each strain for 2 months
LGG, *B. infantis*, *B. breve*, *L. bulgaricus*, *Streptococcus thermophilus*, *L. acidophilus*, and *B. bifidum*	10^9^ CFU total dose for 3 months

CFU: colony-forming unit. 2×: twice daily. Adapted from Williams et al. [103].

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
