# Peer review of "The Interplay Between Nutrition and Microbiota and the Role of Probiotics and Symbiotics in Pediatric Infectious Diseases"

_nutrients, 2025, doi:10.3390/nu17071222_

Round 1

Reviewer 1 Report

Comments and Suggestions for Authors

The manuscript effectively underscores the critical intersection between nutrition and infectious diseases in pediatric health, particularly within immunocompromised and critically ill populations.

Here are my comments: 

1] The topic is highly engaging; however, the inclusion of figures illustrating metabolic pathways would significantly enhance comprehension and clarity.

2] To boost coherence, clinical studies should be presented in a separate section, allowing for a more structured and focused analysis of their findings and implications.

3] Sections 5 and 6 lack sufficient textual content, limiting the depth of analysis and discussion. Expanding these sections would provide a more comprehensive evaluation of the topic. Additionally, the abstract requires expansion (around 250 words) to provide a more thorough summary of the review's objectives.

4] It is recommended to include a table delineating the optimal concentrations of probiotics and prebiotics that should be present in the human organism for promoting health and maintaining a balanced microbiome.

Reviewer 2 Report

Comments and Suggestions for Authors

This review-type manuscript focuses on a controversial topic in the literature – the interplay between gut microbiota-nutrition-infections disease in the pediatric population. In my opinion, it is a well-organized manuscript regarding this particular subject. To increase the quality of the manuscript, I have a few suggestions for the authors:
-    Kindly add two images in subchapters 2 and 3 to highlight the illustrated data. 
-    Please add the limits of this study at the end of Section 5.
-    Kindly organize an additional subsection in the main text to explore the relationship between malnutrition-gut microbiota and infectious diseases in the pediatric population.
-    Kindly add a reference in lines 38, 77, 124, 149, 233, 304, and 333
-    Please revise the reference list according to the recommendations of the Nutrients journal.

Author Response

Please see the attached file. Thank you

We have been experiencing several difficulties with the reference citation editor and the program keeps changing the reference list once corrected. If necessary, we can provide the reference list in a separate file. We apologize for any inconvenience this may cause.

Round 2

Reviewer 1 Report

Comments and Suggestions for Authors

Line 49: remove the parenthesis.

The legend of Figure 1 should be placed at the bottom.

Author Response

Both changes have been made in the revised manuscript. Thanks again for your revision of our paper.

Reviewer 2 Report

Comments and Suggestions for Authors

The manuscript has been improved by responding to comments and integrating missing references.

However, I have only a few suggestions for the authors:

  • Kindly increase the typographical quality of the Figure 1.
  • As I previously mentioned, revise the reference list according to the recommendations of the Nutrients journal”.

https://www.mdpi.com/journal/nutrients/instructions

Author Response

Thanks again for your revision. We have tried to improve Figure 1 typographical quality and the reference list. We provide the reference list also in a separate file.
